# Metathesis of Functionalized Alkane: Understanding the Unsolved Story

**Mykyta Tretiakov, Yury Lebedev, Manoja K. Samantaray \*, Aya Saidi, Magnus Rueping \* and Jean-Marie Basset \***

KAUST Catalysis Center (KCC), King Abdullah University of Science & Technology, Thuwal 23955-6900, Saudi Arabia; mykyta.tretiakov@kaust.edu.sa (M.T.); yury.lebedev@kaust.edu.sa (Y.L.); aya.saidi@kaust.edu.sa (A.S.)

\* Correspondence: manoja.samantaray@kaust.edu.sa (M.K.S.); magnus.rueping@kaust.edu.sa (M.R.); jeanmarie.basset@kaust.edu.sa (J.-M.B.)

**Abstract:** For the first time, we developed a method which enables a functionalized alkane to be metathesized to its lower and higher homologues. For this metathesis reaction, we used $[(\equiv Si\text{-}O\text{-})W(CH_3)_5]$ as a catalyst precursor and 9-hexyl-9*H*-carbazole as a reactant.

**Keywords:** metathesis; functionalized alkane; tungsten; carbazole; reaction mechanism; pyrrole

## 1. Introduction

Since the discovery of alkane metathesis reaction by well-defined silica supported [Ta]-H catalyst [1], many groups, including us, have been continuously working on the development of well-defined catalysts for "functionalized" alkanes metathesis reaction or alkyl chains bearing any functionality, along with improved reactivity in alkane metathesis reaction [2–8]. While significant progress was achieved in improving the catalyst activity (e.g., by changing catalyst support and by employing a tandem system, where two catalysts act back to back for cascade reaction), metathesis of a functionalized alkane remains elusive in the scientific community [7,9–12]. To metathesize a functionalized alkane, the reactant should undergo dehydrogenation to a functionalized olefin, functionalized olefin metathesis, and finally reduction of the newly-formed, functionalized olefins to new functionalized alkanes. The main difficulty in functionalized alkane metathesis reaction is the functional moiety, which is expected to poison the catalyst by coordinating with the metal, and hence deactivating the system for further reaction. Therefore, it is a tremendous challenge to metathesize an alkane having a functional group. Herein we present the first example of functionalized alkane metathesis using silica supported $W(CH_3)_5$ **1** as a catalyst precursor and 9-Hexyl-9*H*-carbazole **6** as a reactant.

## 2. Results and Discussion

From previous publications it was known that ligands that have a lone pair of electrons coordinate with electron-deficient early-transition metals, making them electron-rich and blocking the coordination site for C-H bond activation [13]. As the metal is more electron-rich and coordinatively more saturated, its affinity towards alkane decreases [14]. Considering, that alkane metathesis occurs first by sigma-bond metathesis with a $d^0$ system, this becomes increasingly important [15].

To avoid this problem, while carrying out the functionalized alkane metathesis reaction, our first aim was to focus on the protection of the functional group. To implement this idea, we chose *N*-Alkyl pyrroles, with the assumption that the lone pair of electrons on the nitrogen atom is involved in the formation of an aromatic system and thus is poorly available for coordination. To our surprise, after many attempts, we only observed a starting material along with its decomposition products.

A literature report shows that pyrroles interact with Lewis acids to produce pyrrole oligomers through the activation of its α position [16]. To avoid this, and to carry out the reaction, we thought to protect the most reactive α position of the pyrrole ring by methyl groups, in order to avoid unwanted reactions of the pyrrole ring while carrying out the metathesis. We synthesized 1-Hexyl-2,5-Dimethyl-1*H*-pyrrole **2** and 1-Propyl-2,5-Dimethyl-1*H*-pyrrole **3** [17,18] (Scheme 1) and carried out the metathesis reaction using [(≡Si-O-)W(CH$_3$)$_5$] **1** as a catalyst precursor (Figure 1).

**1**

monopodal species
with silica at 700°C

**Figure 1.** Well-defined silica-supported [(≡Si-O-)W(CH$_3$)$_5$] **1.**

$$R = \begin{array}{ll} \text{Hexyl} & \textbf{2} \\ \text{Propyl} & \textbf{3} \\ \text{Allyl} & \textbf{4} \end{array}$$

**Scheme 1.** Synthesis of *N*-Alkyl pyrroles.

It has been already observed that **1** leads to [(≡Si-O-)W(H)$_3$(=CH$_2$)] during catalyst activation [19]. While testing this family of molecules, expecting the metathesis products of the reactants **2** and **3**, we ended up with unreacted starting material along with traces of decomposition, and isomerization of both the substrates without any metathesis product.

The alkane metathesis consists of mainly three subsequent paths—dehydrogenation, olefin metathesis, and hydrogenation—as we did not observe any expected metathesis products from the above two reactants, we decided to investigate if the olefin metathesis step could be performed in our catalytic system. We synthesized 1-Allyl-2,5-Dimethyl-1*H*-pyrrole **4**, which is the olefinic analogue of the 1-Propyl-2,5-Dimethyl-1*H*-pyrrole **2**, for olefin metathesis reaction using catalyst **1** (Figure 2).

The product was analyzed by gas chromatography (GC), gas chromatography coupled with mass spectrometry (GC-MS), and nuclear magnetic resonance (NMR) spectroscopy, confirming the formation of expected product 1,4-bis-(2,5-Dimethyl-1*H*-pyrrole-1yl)but-2-ene, with 85% conversion and a turnover number (TON) of 850 (Figure 2) [20–22]. These results clearly indicated that our catalyst was capable of carrying out the olefin metathesis reaction for this particular substrate, whereas in the case of 1-Propyl-2,5-Dimethyl-1*H*-pyrrole **2** or 1-Hexyl-2,5-Dimethyl-1*H*-pyrrole **3**, no metathesis was observed. The limiting step in the latter case was the alkyl chain C-H activation (as an entry into the alkane metathesis productive cycle). During the catalytic reaction, we understood that isomerization of the substrates occurs because of walking of α-methyl groups on the ring [23]. To avoid that, we thought to synthesize 9-Hexyl-9*H*-carbazole (a rigid molecule) to carry out the metathesis reaction. 9-Hexyl-9*H*-carbazole was obtained by the reaction of carbazole and hexyl bromide in the presence of KOH, by following a literature procedure (Scheme 2) [24].

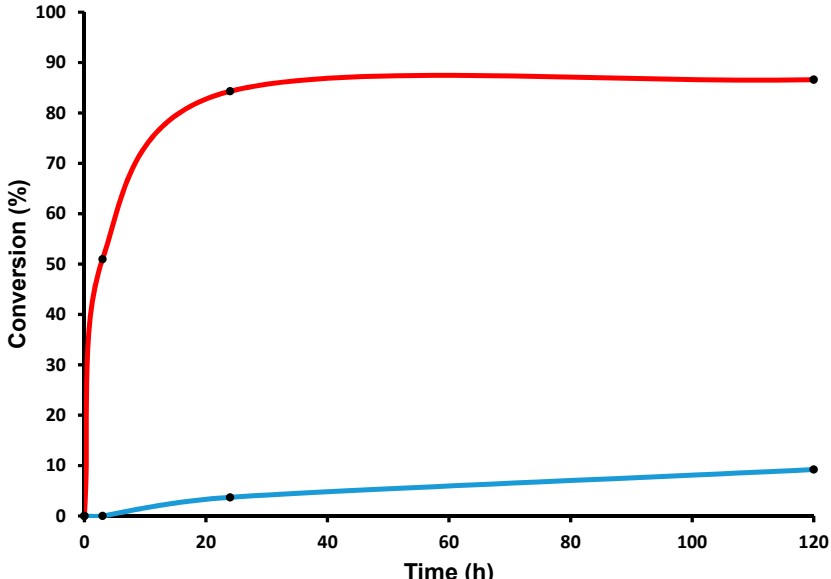

**Figure 2.** Metathesis of 1-Alkyl-2,5-Dimethyl-1*H*-pyrrole, conversion versus time plot. (■) Reaction at room temperature (RT); (■) reaction at 150 °C.

**Scheme 2.** Synthesis of 9-Hexyl-9*H*-carbazole **6.**

In a typical reaction, 9-Hexyl-9*H*-carbazole **6** and [(≡Si-O-)W(CH$_3$)$_5$] **1**, in 50:1 molar ratio, based on loading of W, were mixed. The mixture was sealed in an ampoule tube under high vacuum and the reaction continued at 150 °C. The first sample was quenched with dichloromethane and injected in GC after one day of reaction. To our surprise, neither any considerable amount of product nor decomposition of the starting material (9-Hexyl-9*H*-carbazole) were observed. We continued the reaction for another four days. After five days, the reaction mixture was taken, quenched with dichloromethane, filtered, and analyzed by GC. We could observe a range of alkylated carbazoles, starting from cC$_2$ to cC$_{10}$, with a conversion of starting material of 2.5% and the formation of carbazole itself (Figures 3 and 4). While carrying out the reaction with **1**, we thought the reaction would go smoother, as the lone pair was delocalized in the carbazole conjugated system and there was no labile group on the ring. To our surprise, we only obtained 5.0% conversion, even after 30 days. While analyzing the reaction mixture, we observed linear alkanes (alkane metathesis products), *N*-Alkyl carbazoles and the dimer of 9-Hexyl-9*H*-carbazole (functionalized alkane metathesis products) (Figures 3 and 4; Supplementary Material Figures S15–S17), along with a considerable amount of carbazole.

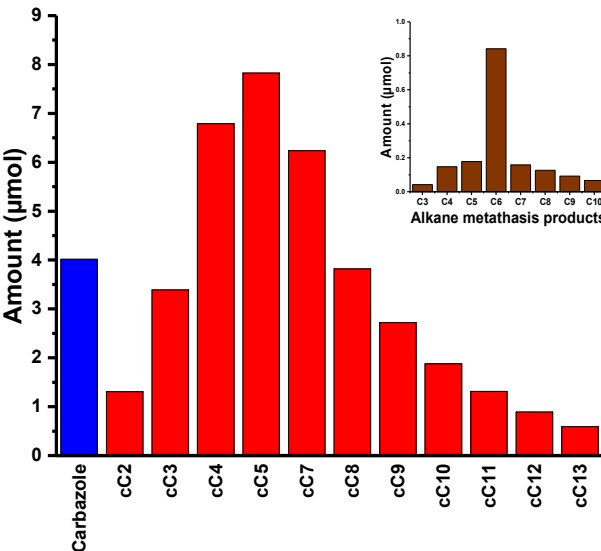

**Figure 3.** Product distribution: Functionalized alkane metathesis products (■), alkane metathesis products (■), and carbazole (■) after 30 days of reaction.

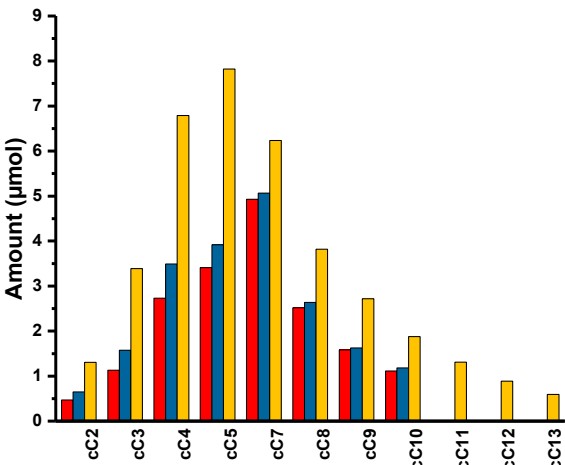

**Figure 4.** Product distribution: Functionalized alkane metathesis products over time (■) 5 days; (■) 10 days; (■) 30 days.

The formation of the dimer [1,10-di(9*H*-carbazole-9-yl-)decane] was understandable as it is the self-metathesis product of the 9-(hex-5en-1-yl)-9*H* carbazole (Supplementary Material Figures S15 and S16). Initially, 9-(hex-5en-1-yl)-9*H* carbazole (dehydrogenation product of 9-Hexyl-9*H*-carbazole) was formed by the reaction of [(≡Si-O-)W(CH$_3$)$_5$] **1** and 9-Hexyl-9*H*-carbazole (Figures S13 and S14). Other *N*-Alkyl carbazoles and alkanes were formed due to chain walking (our active catalyst contains W-H) followed by cross metathesis with linear olefins, and, at the end, reduction of newly formed olefins to new alkanes. Looking at the product distribution, we observed that, along with all the metathesis products of the functionalized alkanes, a considerable amount of *N*-alkane metathesis products and carbazole were formed. In *N*-alkane metathesis products, the concentration of the C$_6$ (hexane) was higher, as compared to its lower and higher homologues. These products could only form if the catalyst attacks the position 1 of the 9-Hexyl-9*H*-carbazole (Scheme 2 and Figure 5), forming the W-C bond followed by reduction of the N-C bond to generate carbazole and the W-Alkyl chain (Figure 5). Furthermore, the W-Alkyl chain underwent an α-H abstraction and β-H elimination, generating W-carbene and olefin. Additionally, W-carbene and olefin underwent an olefin metathesis reaction, generating a range of *N*-alkane products (C$_3$–C$_{10}$), after reduction of the newly-formed olefins (Figure 5). The various *N*-Alkyl carbazoles were formed by activation of position 6 (Scheme 2) of the

alkyl chain of the 9-Hexyl-9*H*-carbazole. Even though we metathesized a functionalized alkane, we could only achieve 5.0% conversion with a TON of **2**. We believe the low conversion was due to the poisoning of the catalyst by the formation of carbazole (Figure 5) during the reaction (since the N-H of carbazole was no longer protected, it could attack the catalyst and deactivate it for further reaction).

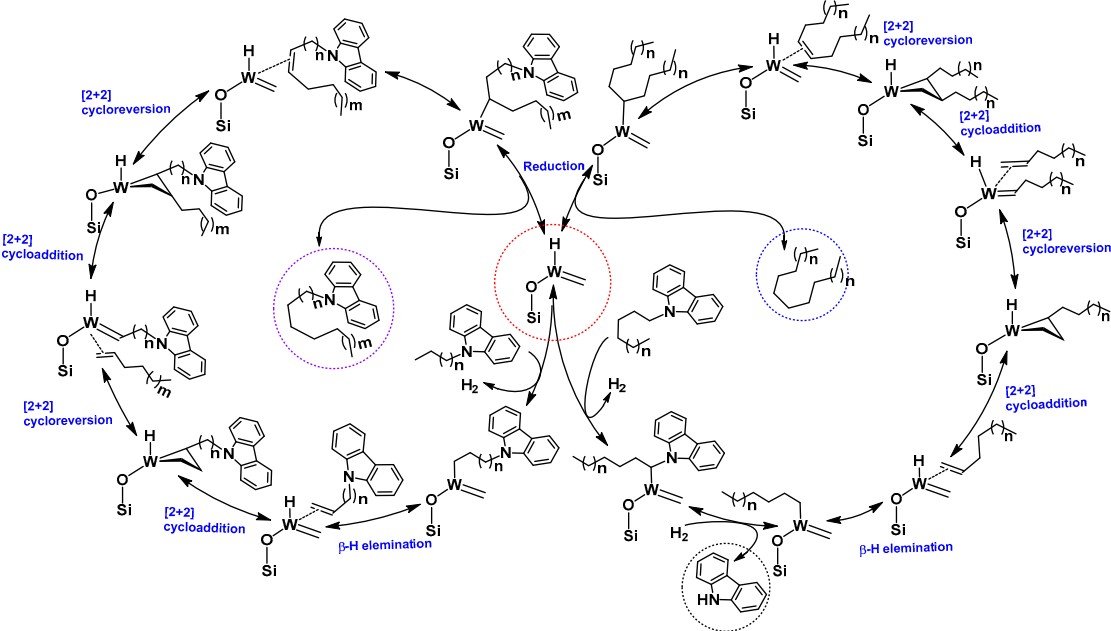

**Figure 5.** Proposed mechanism for the functionalized alkane metathesis reaction.

## 3. Conclusions

For the first time, we carried out a functionalized alkane metathesis reaction using silica-supported catalyst [(≡Si-O-)W(CH$_3$)$_5$] **1**, with a conversion of 5.0%; *N*-alkane metathesis was also observed due to the reaction course. To avoid catalyst decomposition, we chose 9-Hexyl-9-*H*-carbazole, whereby the lone pair was delocalized in the conjugated system. Our study shows that the catalyst decomposes because of the reaction of the N-H of the carbazole with the catalyst. Currently, we are focusing on the development of a robust catalyst for the functionalized alkane metathesis reaction.

## 4. Materials and Methods

### 4.1. General Experimental Procedures

All experiments were carried out by using standard Schlenk and glove box techniques under an inert nitrogen atmosphere. The syntheses and the treatments of the surface species were carried out using high vacuum lines (<10$^{-5}$ mbar) and glove-box techniques. Pentane was distilled from a Na/K alloy under N$_2$ and dichloromethane from CaH$_2$. Both solvents were degassed through freeze-pump-thaw cycles. SiO$_{2-700}$ was prepared from Aerosil silica from Degussa (Frankfurt, Germany) (specific area of 200 m$^2$/g), which were partly dehydroxylated at 700 °C under high vacuum (<10$^{-5}$ mbar) for 24 h to give a white solid, having a specific surface area of 190 m$^2$/g and containing around 0.5–0.7 OH/nm$^2$. W(CH$_3$)$_6$ (Figures S1–S3) and supported pre-catalyst **1** were prepared according to literature procedures (Figure S4) [25–27]. 2,5-hexanedione, hexylbromide, hexylamine, propylamine, allylamine, carbazole, C$_2$-C$_{13}$-alkylbromides, and 1,10-dibromodecane were purchased from Aldrich.

NMR spectra were recorded on Bruker Avance III 500 MHz NMR spectrometer (Bruker Biospin AG, Fallanden, Switzerland) equipped with a BBFO CryoProbe (Bruker). IR spectra were recorded on a Nicolet 6700 FT-IR spectrometer (Thermo Fisher Scientific, Waltham, MA, USA) by using a

DRIFT cell equipped with CaF$_2$ windows (Figure S4). The IR samples were prepared under argon within a glove box. Typically, 64 scans were accumulated for each spectrum (resolution 4 cm$^{-1}$). GC measurements were performed with an Agilent 7890A Series (FID detection; Santa Clara, CA, USA). Method for GC analyses: Column HP-5; 30 m length × 0.32 mm ID × 0.25 μm film thickness; flow rate, 1 mL/min (N$_2$); split ratio, 50/1; inlet temperature, 250 °C, detector temperature, 250 °C; temperature program, 40 °C (3 min), 40–250 °C (12 °C/min), 250 °C (3 min), 250–300 °C (10 °C/min), 300 °C (3 min); 9-hexyl-9*H*-carbazole retention time, t$_R$ = 21.67 min. GC-MS measurements were performed with an Agilent 7890A Series coupled with an Agilent 5975C Series. A GC-MS, equipped with a capillary column coated with none polar stationary phase HP-5MS, was used for molecular weight determination and identification, which allowed the separation of chemical compounds according to their boiling point differences. Method for GC-MS analyses: Column HP-5MS; 30 m length × 0.25 mm ID × 0.25 μm film thickness; flow rate, 1 mL/min (N$_2$); split ratio, 100/1; inlet temperature, 220 °C, detector temperature, 300 °C; temperature program, 150 °C (0 min), 150–300 °C (20 °C/min), 300 °C (30 min).

HPLC-MS analysis was performed at Core Lab ACL KAUST. The separation of the 1,10-di(9*H*-carbazole-9yl-)decane from the metathesis reaction mixture was performed using an Accela HPLC System and a hypersil gold 2.1 × 50 mm (Thermo Fisher Scientific, Waltham, MA, USA). 1 mg of the sample was dissolved in 1 mL (methanol:chloroform), the liquid compound was then diluted ×1000 in 1 mL of methanol. The separation was achieved using a gradient composed of water and methanol. The mobile phase solvents were composed of (a) 100% water plus 0.1% formic acid, and (b) 100% methanol plus 0.1% formic acid. The gradient elution program is summarized in Table S1. The injection volume was 10 μL. The flow rate was 400 μL/min. The data processing was performed using Xclaibur Software (Thermo Fisher Scientific). The mass analysis was performed using a Thermo LTQ Velos Orbitrap mass spectrometer (Thermo Scientific, Pittsburgh, PA, USA) equipped with a heated electron spray ionization (ESI) ion source. The mass scan range was set to 100–1000 *m/z*, with a resolving power of 100,000. The *m/z* calibration of the LTQ-Orbitrap analyzer was performed in the positive ESI mode, using a solution containing caffeine, MRFA (met-arg-phe-ala) peptide, and Ultramark 1621 according to the manufacturer's guidelines. The ESI was performed with a heated ion source equipped with a metal needle and operated at 4 kV. The source vaporizer temperature was adjusted to 350 °C, the capillary temperature was set at 250 °C, and the sheath and auxiliary gases were optimized and set to 40 and 20 arbitrary units.

### 4.2. Synthesis of N-Hexyl-Pyrrole

*N*-Hexyl-pyrrole was synthesized according to literature procedure [17,18].

### 4.3. Synthesis of N-Hexyl-2,5-Dimethyl-Pyrrole 2, N-Propyl-2,5-Dimethyl-Pyrrole 3 and N-Allyl-2,5-Dimethyl-Pyrrole 4

Compounds **2**, **3**, **4** were synthesized according to the Paal-Knorr method of pyrrole synthesis (Scheme 3).

**Scheme 3.** Synthesis of *N*-alkyl pyrroles.

- General Procedure for **2**, **3**, **4** (Figures S5–S10):

2,5-hexanedione (10 mmol) was dissolved in MeOH (50 mL) and the respective primary amine (10 mmol) was added at room temperature (RT). The reaction was slightly exothermic and the cooling by ice bath was recommended when higher quantities were used for reaction. Reaction mixture (RM) was stirred for 10 min at RT and evaporated in vacuum. Hexane was added to the RM and the water phase separated from the organic phase. The organic phase was dried over MgSO₄, filtered, and evaporated to dryness. The product was purified by flash-column chromatography (Hexane:EtOAc = 4:1), distilled over sodium in vacuum, and degassed prior to use. The reference *N*-Alkyl-2,5-Dimethyl-pyrroles (C2–C9) were synthesized with the same procedure and used as standards in GC-MS analysis. Yields were in range 60–70%.

*N-Hexyl-2,5-Dimethyl-pyrrole* **2**. $^1$H NMR ($C_6D_6$), δ, ppm (*J*, Hz): 6.04 (2H, s, CH Ar), 3.35–3.32 (2H, t, *J* = 7.2, N-CH₂), 2.09 (6H, s, 2 × Ar CH₃), 1.36–1.30 (2H, m, CH₂), 1.18–1.14 (2H, m, CH₂), 1.07–1.04 (4H, m, 2 × CH₂), 0.85–0.82 (3H, t, *J* = 7.2, CH₃). $^{13}$C NMR ($C_6D_6$): 126.7, 106.2, 43.5, 31.8, 31.4, 26.8, 22.9, 14.2, 12.7.

*N-Propyl-2,5-Dimethyl-pyrrole* **3**. $^1$H NMR ($C_6D_6$), δ, ppm (*J*, Hz): 6.02 (2H, s, CH Ar), 3.27–3.24 (2H, t, *J* = 7.5, N-CH₂), 2.05 (6H, s, 2 × Ar CH₃), 1.34–1.27 (2H, m, *J* = 7.4, CH₂), 0.63–0.60 (3H, t, *J* = 7.3, CH₃). $^{13}$C NMR ($C_6D_6$): 126.8, 106.1, 44.9, 24.5, 12.7, 11.2.

*N-Allyl-2,5-Dimethyl-pyrrole* **4**. $^1$H NMR ($C_6D_6$), δ, ppm (*J*, Hz): 6.03 (2H, s, CH Ar), 5.50–5.43 (1H, m, CH Allyl), 4.85–4.82 (1H, dd, $^1J$ = 10.3, $^2J$ = 1.8 Hz), 4.57–4.53 (1H, dd, $^1J$ = 10.3, $^2J$ = 1.8 Hz), 3.82–3.80 (2H, m, CH₂), 2.01 (6H, s, 2 × CH₃). $^{13}$C NMR ($C_6D_6$): 134.8, 127.1, 114.9, 106.1, 45.2, 12.3.

### 4.4. 1,4-Bis(2,5-Dimethyl-1-H-Pyrrole-1yl)But-2-Ene

1,4-bis(2,5-dimethyl-1-*H*-pyrrole-1yl)but-2-ene was separated from *N*-Allyl-2,5-Dimethyl-pyrrole metathesis reaction mixture as a white solid and characterized by NMR (Figures S11 and S12).

$^1$H NMR ($C_6D_6$), δ, ppm (*J*, Hz): 6.04 (4H, s, CH Ar), 4.70–4.69 (2H, t, *J* = 1.2 Hz, 2 × CH), 3.59 (4H, d, *J* = 1 Hz, 2 × CH₂), 1.95 (12H, s, CH₃). $^{13}$C NMR ($C_6D_6$): 127.0, 126.9, 106.1, 43.8, 12.3.

### 4.5. Synthesis of 9-Hexyl-9H-Carbazole 6

9-Hexyl-9*H*-carbazole **6** was synthesized according to the literature method with a modification [24].

Procedure: carbazole (37 mmol) was dissolved in DMF (85 mL) under argon, potassium hydroxide (232 mmol) was added to a solution, and the mixture was stirred for 40 min at RT. Hexylbromide (37 mmol) was added dropwise and the RM was stirred for an additional 9 h at RT. After that, RM was poured into water (100 mL). In order to extract the product from the DMF/water mixture, 100 mL of n-hexane and 100 mL of EtOAc were added. The resulting four-component (DMF/water/hexane/EtOAc) mixture was intensively shacked in the separation funnel. The upper organic phase was separated and washed with water (2 × 50 mL). The lower DMF/water phase was additionally extracted by a mixture of hexane (100 mL) and EtOAc (100 mL), and the organic phase was washed with water (2 × 50 mL). The combined organic phase (pure from DMF) was dried over Na₂SO₄ and evaporated in vacuum. The product was purified by column chromatography (petroleum ether:DCM = 3:1), recrystallized from hexane (−30 °C) and dried on high-vacuum line HVL. Yield: 86%, white needle-like crystals. NMR spectra were in accordance with the literature data (Figures S13 and S14).

$^1$H NMR ($C_6D_6$), δ, ppm (*J*, Hz): 8.08 (2H, d, *J* = 7.8 Hz, H Ar), 7.44–7.40 (2H, m, H Ar), 7.25–7.22 (2H, m, H Ar), 7.20–7.18 (2H, m, H Ar), 3.79 (2H, t, *J* = 7.2 Hz, CH₂), 1.51–1.48 (2H, m, CH₂), 1.08–1.02 (6H, m, 3 × CH₂), 0.79–0.76 (3H, t, *J* = 7 Hz, CH₃). $^{13}$C NMR ($C_6D_6$): 140.9, 125.9, 123.5, 120.8, 119.2, 109.0, 42.9, 31.8, 29.0, 27.1, 22.8, 14.2.

### 4.6. Synthesis of 9-Hexyl-9H-Carbazole Metathesis Products

The metathesis products (*N*-Alkyl-carbazoles, $C_2$–$C_{13}$) were additionally prepared as reference compounds, according to the procedure from Section 4.5. and used for GC-MS analysis without further purification. The retention time and fragmentation of the reference compounds exactly matched the products from the 9-Hexyl-9*H*-carbazole metathesis reaction, proving their linear structure. Retention times for *N*-alkylated carbazoles: $C_2$ 4.73 min, $C_3$ 5.11 min, $C_4$ 5.55 min, $C_5$ 5.98 min, $C_6$ (starting material) 6.51 min, $C_7$ 6.87 min, $C_8$ 7.27 min, $C_9$ 7.67 min, $C_{10}$ 8.13 min, $C_{11}$ 8.63 min, $C_{12}$ 9.20 min, and $C_{13}$ 9.86 min.

### 4.7. 1,10-Di-(9H-Carbazole-9yl-)-Decane (C$_{34}$H$_{37}$N$_2$)

1,10-di-(9*H*-carbazole-9yl-)-decane was synthesized as a reference compound, according to the procedure from Section 4.5., with 1,10-dibromodecane as an alkylation agent, and used for HPLC-MS analysis of the reaction mixture (Figures S15–S17).

$^1$H NMR ($C_6D_6$), δ, ppm (*J*, Hz): 8.07 (4H, d, *J* = 7.7, H Ar), 7.43–7.40 (4H, m, H Ar), 7.24–7.19 (8H, m, H Ar), 3.81 (4H, t, *J* = 7, 2 × N-CH$_2$), 1.54–1.48 (4H, m, 2 × CH$_2$), 1.05–0.95 (12H, m, 6 × CH$_2$). $^{13}$C NMR ($C_6D_6$): 140.9, 125.9, 123.5, 120.9, 119.3, 109.0, 42.9, 29.6, 29.1, 27.4.

Theoretically predicted isotopic distribution of $C_{34}H_{37}N_2$: 473.29513, 474.29848, 475.30184 (δ = 0 ppm). Experimentally found isotopic distribution of the reference compound: 473.29497, 474.29824, 475.30157 (δ = −0.335 ppm). Experimentally-found isotopic distribution of the dimer found in the reaction mixture: 473.29506, 474.29837, 475.30188 (δ = −0.147 ppm).

### 4.8. Catalytic Reactions

9-Hexyl-9*H*-carbazole **6** (200 mg, 0.8 mmol) and [(≡Si-O-)W(CH$_3$)$_5$] **1** (100 mg, 0.016 mmol W) were mixed in the ampule tube in the glove box; the ampule tube was connected to a high-vacuum line and sealed in a vacuum. The ampule was then heated at 150 °C with stirring (50 rpm). Quenching: the ampule was cooled down in liquid nitrogen, opened, 1 mL of DCM was added, the mixture was filtered through a syringe filter (1 μm), and the sample was analyzed by GC.

1-Hexyl-2,5-Dimethyl-1*H*-pyrrole **2**, 1-Propyl-2,5-Dimethyl-1*H*-pyrrole **3**, and 1-Allyl-2,5-Dimethyl-1*H*-pyrrole **4** were analyzed with the same procedure.

**Supplementary Materials:** The following are available online at http://www.mdpi.com/2073-4344/9/3/238/s1, Table S1: Gradient elution program, Figure S1: $^1$H NMR spectrum of WMe$_6$ in CD$_2$Cl$_2$ at 203 K, Figure S2: $^{13}$C NMR spectrum of WMe$_6$ in CD$_2$Cl$_2$ at 203 K, Figure S3: $^2$D solution $^1$H-$^{13}$C Heteronuclear Single Quantum Correlation (HSQC) NMR spectrum of WMe$_6$ in CD$_2$Cl$_2$ at 203 K, Figure S4: FT-IR spectroscopy of silica partially dehydroxylated at 700 ºC (SiO$_{2-700}$) (blue curve) and W(CH$_3$)$_6$ grafted on SiO$_{2-700}$ (**1**) (orange curve), Figure S5: $^1$H NMR spectrum of *N*-Hexyl-2,5-Dimethyl-pyrrole **2** in C$_6$D$_6$, Figure S6: $^{13}$C NMR spectrum of *N*-Hexyl-2,5-Dimethyl-pyrrole **2** in C$_6$D$_6$, Figure S7: $^1$H NMR spectrum of *N*-Propyl-2,5-Dimethyl-pyrrole **3** in C$_6$D$_6$, Figure S8: $^{13}$C NMR spectrum of *N*-Propyl-2,5-Dimethyl-pyrrole **3** in C$_6$D$_6$, Scheme S1: No functionalized alkane metathesis product was observed using substrate **2** and **3**, Figure S9: $^1$H NMR spectrum of *N*-Allyl-2,5-Dimethyl-pyrrole **4** in C$_6$D$_6$, Figure S10: $^{13}$C NMR spectrum of *N*-Allyl-2,5-Dimethyl-pyrrole **4** in C$_6$D$_6$, Figure S11: $^1$H NMR spectrum of 1,4-bis(2,5-dimethyl 1-*H*-pyrrole-1yl)but-2-ene in C$_6$D$_6$, Figure S12: $^{13}$C NMR spectrum of 1,4-bis(2,5-dimethyl 1-*H*-pyrrole-1yl)but-2-ene in C$_6$D$_6$, Figure S13: $^1$H NMR spectrum of 9-Hexyl-9*H*-carbazole in C$_6$D$_6$, Figure S14: $^{13}$C NMR spectrum of 9-Hexyl-9*H*-carbazole in C$_6$D$_6$, Figure S15: $^1$H NMR spectrum of 1,10-di(9*H*-carbazole-9yl-)decane in C$_6$D$_6$, Figure S16: $^{13}$C NMR spectrum of 1,10-di(9*H*-Carbazole-9yl-)decane in C$_6$D$_6$, Figure S17: Comparison of the reference 1,10-di-(9*H*-carbazole-9yl-)-decane (C$_{34}$H$_{37}$N$_2$) with the dimer found in 9-Hexyl-9*H*-carbazole metathesis reaction mixture.

**Author Contributions:** Conceptualization, M.T., Y.L., M.K.S., M.R. and J.-M.B.; experiments, M.T., Y.L., A.S.; writing M.T., M.K.S., and J.-M.B.

**Funding:** This work was supported by funds from King Abdullah University of Science and Technology (KAUST).

**Acknowledgments:** The authors acknowledge Salim Sioud (KAUST ACL Core Lab), Kristina Smirnova and Miguel Dinis Veloso Guerreiro for technical assistance.

**Conflicts of Interest:** The authors declare no competing financial interests.

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
