# Peer review of "Metathesis of Functionalized Alkane: Understanding the Unsolved Story"

_catalysts, doi:10.3390/catal9030238_

Round 1

Reviewer 1 Report

This manuscript describes a challenge for the alkane metathesis of functionalized alkanes. Silica-supported {Ta]-H or W(CH3)5 can be used for an alkane metathesis catalyst. As mentioned by authors, however, functionalized alkanes having a coordinating group to metal centers such as amino groups are not suitable reagents for this reaction. To get over this problem, author's proposal seems to be reasonable. Although the results described in this manuscript show problematic further reaction, these 

information will give researchers understanding to achieve the alkane metathesis using functionalized alkanes. 

Thus, the reviewer would recommend that this review is suitable for publication in Molecules.

Please correspond the below points.

1) In the last part of the Introduction, is the word "silica supported W(CH3)6" correct? W(CH3)6 is the homogeneous form and after supported on silica W(CH3)5 is correct unit as shown in Figure 1, the reviewer thinks.

2) In page 2, line 6, 1-propyl-2,5-dimethyl-1H-pyrrole is assigned to "2" and 1-hexyl-2,5-dimethyl-1H-pyrrole is "3". But, other parts in the manuscript and SI listed that hexyl substrate is "2" and propyl is "3", e.g. Scheme 1.

3) In page 2, the first sentence in second paragraph, "1" is not written as bold number.

4) Is there two spaces between "consists of" and "mainly"?

5) In the last part of page 3, Figure 3,5 seems not to be correct. Is "Figures 3,4" correct?

6) In the explanation of the result of the reaction using "6", authors showed that the dimer of 9-hexyl-9H-carbazole (functionalized alkane metathesis products) was observed. Actually, Figure S17 demonstrated the generation of this dimer. However, the yield of this product was not included in the Figures 3 and 4. How amount was this product generated?

Author Response

Comment to the author

This manuscript describes a challenge for the alkane metathesis of functionalized alkanes. Silica-supported {Ta]-H or W(CH3)5 can be used for an alkane metathesis catalyst. As mentioned by authors, however, functionalized alkanes having a coordinating group to metal centers such as amino groups are not suitable reagents for this reaction. To get over this problem, author's proposal seems to be reasonable. Although the results described in this manuscript show problematic further reaction, these information will give researchers understanding to achieve the alkane metathesis using functionalized alkanes. 

Thus, the reviewer would recommend that this review is suitable for publication in Molecules.

We thank the reviewer for appreciating our work and recommending for a possible publication. 

Comment to the author

Please correspond the below points.

 In the last part of the Introduction, is the word "silica supported W(CH3)6" correct? W(CH3)6 is the homogeneous form and after supported on silica W(CH3)5 is correct unit as shown in Figure 1, the reviewer thinks.

We changed from W(CH3)6 to W(CH3)5 in the main text (Please see the last line of  1st paragraph page 1)

Comment to the author

In page 2, line 6, 1-propyl-2,5-dimethyl-1H-pyrrole is assigned to "2" and 1-hexyl-2,5-dimethyl-1H-pyrrole is "3". But, other parts in the manuscript and SI listed that hexyl substrate is "2" and propyl is "3", e.g. Scheme 1.

We changed the name accordingly as 1-hexyl-2,5-dimethyl-1H-pyrrole 2 and 1-propyl-2,5-dimethyl-1H-pyrrole 3, please see main text page page 2 1st paragraph. 

Comment to the author

In page 2, the first sentence in second paragraph, "1" is not written as bold number.

We marked 1 as bold in the paragraph. Please see the main text page 2 second paragraph.    

Comment to the author

Is there two spaces between "consists of" and "mainly"?

We modified it according to the reviewer.

Comment to the author

In the last part of page 3, Figure 3,5 seems not to be correct. Is "Figures 3,4" correct?

We changed the figure number accordingly. Please see main text pages 3 and 4.

Comment to the author

In the explanation of the result of the reaction using "6", authors showed that the dimer of 9-hexyl-9H-carbazole (functionalized alkane metathesis products) was observed. Actually, Figure S17 demonstrated the generation of this dimer. However, the yield of this product was not included in the Figures 3 and 4. How amount was this product generated?

The dimer was only observed by HPLC-MS and was not isolated as the amount was very small. To prove that the little amount is the dimer, we separately synthesized the compound and carried out MS to match it with the metathesis product.  

Reviewer 2 Report

The authors presented their findings in a clear and concise manner, thereby drawing the interest of readers to their research. They showed that by first protecting the functional group of the functionalized alkane, it is possible to carry out alkane metathesis because poisoning of the catalyst by the functional group can then be avoided. It is an interesting method as a starting point, but they concluded that poisoning of catalyst still occurred due to the formation of carbazole. Although the conclusion/ findings need improvement, this work is still interesting as a proof of concept.

It is mentioned in the introduction that 'many groups, including us, were continuously working on the development of well-defined catalysts...' and there were 10 citations in the introduction section. Out of these 10 citations, 7 were self citations and this seems rather excessive. While the authors are experts in this research field as shown by their cited publications in JACS and Science, the authors should include expand their literature to show the developments of other groups as they mentioned.

Author Response

Reviewer-2

Comment to the author

The authors presented their findings in a clear and concise manner, thereby drawing the interest of readers to their research. They showed that by first protecting the functional group of the functionalized alkane, it is possible to carry out alkane metathesis because poisoning of the catalyst by the functional group can then be avoided. It is an interesting method as a starting point, but they concluded that poisoning of catalyst still occurred due to the formation of carbazole. Although the conclusion/ findings need improvement, this work is still interesting as a proof of concept.

We thank the reviewer for appreciating our work and recommending for a possible publication. 

Comment to the author

It is mentioned in the introduction that 'many groups, including us, were continuously working on the development of well-defined catalysts...' and there were 10 citations in the introduction section. Out of these 10 citations, 7 were self citations and this seems rather excessive. While the authors are experts in this research field as shown by their cited publications in JACS and Science, the authors should include expand their literature to show the developments of other groups as they mentioned.

We agree with the reviewer’s comment regarding the citation.  We are pioneer of this field, and from the last 20 years, we are continuously working on the development of the catalyst.  With our citation, we also cite Professor Goldman. According to reviewers’ suggestion, we include another 2 references by Professor R. R. Schrock and by Professor M. Brookhart (Please see reference 11 and 12 of page 1 and page 6).  

Reviewer 3 Report

This paper reported the metathesis of alkyl carbozole catalyzed by supported tungsten complex, which is important to extend the application of metathesis chemistry. The deactivation of active sites was also explored by analyzing the products in the study. Whereas the authors have attempted to address an important research question in catalysis, there are several issues that need to be addressed before the paper is suitable for publication on Catalysts, as listed below:

1.      The labeling of the compound in Scheme 1 needs to be revised. The propyl compound should be labeled as reactant 2 and the hexyl compound should be labeled as reactant 3.

2.      The authors mentioned that species 1 leads to [(≡Si−O−)W(H)3(=CH2)] during catalyst activation on Page 2. Is the procedure of the catalyst activation the same as described in the cited literature (H2 treatment at -78 oC followed by R.T., Reference 17, Chem. Sci. 2016, 7, 1558–1568.)? This needs to be clarified in the supplementary information.

3.      On Page 3, the authors attributed the negligible conversion of 1-propyl-2,5-dimethyl-1H-pyrrole to the isomerization of the substrate, which has been proposed before (Reference 21, Comprehensive Heterocyclic Chemistry, vol. 4, Pergamon Press, Oxford, 1984, p. 201.) It needs to be clarified whether the “walking” of methyl groups (leading to the exposure of α site) or the “walking” of propyl group (leading to the exposure of N-H) causes the poor activity.

4.      On Page 3, the author mentioned that for 9-hexyl-9H-carbazole reaction, negligible conversion was observed after 1day whereas 2.5% conversion was observed after 5 days. Does this indicate the induction period of this process? If so, what causes the induction period?

5.      The deactivation of catalyst during 9-hexyl-9H-carbazole reaction was attributed to the formation of carbazole byproduct (Page 4 and 5). Please clarify how carbazole poison the catalyst. Whether it is caused by the coordination of N to W or by decomposition of W complex? Additionally, is there any feasible method to recycle the catalyst after the deactivation?

Author Response

Reviewer-3

Comment to the author

This paper reported the metathesis of alkyl carbozole catalyzed by supported tungsten complex, which is important to extend the application of metathesis chemistry. The deactivation of active sites was also explored by analyzing the products in the study. Whereas the authors have attempted to address an important research question in catalysis, there are several issues that need to be addressed before the paper is suitable for publication on Catalysts, as listed below:

 We thank the reviewer for appreciating our work

 Comment to the author

The labeling of the compound in Scheme 1 needs to be revised. The propyl compound should be labeled as reactant 2 and the hexyl compound should be labeled as reactant 3.

We revised the names according to the reviewer. The names are 1-hexyl-2,5-dimethyl-1H-pyrrole 2 and 1-propyl-2,5-dimethyl-1H-pyrrole 3. Please see main text page page 2 1st paragraph. 

Comment to the author

The authors mentioned that species 1 leads to [(≡Si−O−)W(H)3(=CH2)] during catalyst activation on Page 2. Is the procedure of the catalyst activation the same as described in the cited literature (H2 treatment at -78 oC followed by R.T., Reference 17, Chem. Sci. 2016, 7, 1558–1568.)? This needs to be clarified in the supplementary information.

In alkane metathesis reaction during the dehydrogenation process H2 evolves, and we believe the evolved H2 reacts with W-CH3 in situ and generate a carbene hydride as we proved it in our previous paper (Chem. Sci. 2016, 7, 1558–1568). 

Comment to the author

On Page 3, the authors attributed the negligible conversion of 1-propyl-2,5-dimethyl-1H-pyrrole to the isomerization of the substrate, which has been proposed before (Reference 21, Comprehensive Heterocyclic Chemistry, vol. 4, Pergamon Press, Oxford, 1984, p. 201.) It needs to be clarified whether the “walking” of methyl groups (leading to the exposure of α site) or the “walking” of propyl group (leading to the exposure of N-H) causes the poor activity.

The walking of the methyl group (attack of the α position).  Please see main text page 3 first paragraph

Comment to the author

On Page 3, the author mentioned that for 9-hexyl-9H-carbazole reaction, negligible conversion was observed after 1day whereas 2.5% conversion was observed after 5 days. Does this indicate the induction period of this process? If so, what causes the induction period?

No, it does not need the induction period, if the induction period is required, then after the induction period the catalyst should perform very well. However, we only observe 2.5 % conversion after 5 days.  We believe, and according to our findings, it generates carbazole during the reaction which deactivates the system. 

Comment to the author

The deactivation of catalyst during 9-hexyl-9H-carbazole reaction was attributed to the formation of carbazole byproduct (Page 4 and 5). Please clarify how carbazole poison the catalyst. Whether it is caused by the coordination of N to W or by decomposition of W complex? Additionally, is there any feasible method to recycle the catalyst after the deactivation?

The carbazole poisons the catalyst either by coordinating to W through lone pair over nitrogen or by forming a direct σ-bond with W by releasing CH4. Once it forms the bond with W, it is not easy to remove the carbazole part from the W and recycle it.